# Self-Compassion and Depressive Symptoms as Determinants of Sensitive Parenting: Associations with Sociodemographic Characteristics in a Sample of Mothers and Toddlers

**DOI:** 10.3390/children10081284

**Published:** 2023-07-26

**Authors:** Bharathi J. Zvara, Sarah A. Keim, Rebecca Andridge, Sarah E. Anderson

**Affiliations:** 1Department of Maternal and Child Health, Gillings School of Public Health, University of North Carolina at Chapel Hill, Chapel Hill, NC 27599, USA; 2Department of Pediatrics, College of Medicine, The Ohio State University, 370 W 9th Ave, Columbus, OH 43210, USA; sarah.keim@nationwidechildrens.org; 3Center for Biobehavioral Health, The Abigail Wexner Research Institute, Nationwide Children’s Hospital, 700 Children’s Drive, Columbus, OH 43205, USA; 4Division of Biostatistics, College of Public Health, The Ohio State University, 1841 Neil Ave, Columbus, OH 43210, USA; andridge.1@osu.edu; 5Division of Epidemiology, College of Public Health, The Ohio State University, 1841 Neil Ave, Columbus, OH 43210, USA; anderson.1767@osu.edu

**Keywords:** self-compassion, sensitive parenting, depressive symptoms, toddler

## Abstract

Parenting that is sensitive and responsive to children’s needs has been shown to support children’s optimal growth and development in many cultural contexts. Numerous studies suggest that self-compassion is positively related to sensitive parenting. Despite growing research interest linking self-compassion to responsive parenting, there are considerable gaps in the literature. The current study examined the associations between self-compassion, depressive symptoms, socioeconomic status, and sensitive parenting. Data was obtained from a cohort study of 300 families in central Ohio enrolled when children were a mean (SD) calendar age of 18.2 (0.7) months. Children of all gestational ages at birth are included, and 37% were born preterm (<37 weeks’ gestation). Observational protocols were used to determine maternal sensitivity in a semi-structured play setting. Self-compassion was assessed with the Self-Compassion Scale when children were 24 months old. Self-compassion was not associated with sociodemographic characteristics including maternal education, household income, child sex and gestational age. In unadjusted regression models, depressive symptoms were related to sensitive parenting (B = −0.036, SE = 0.016, *p* = 0.03), but self-compassion was not a statistically significant predictor (*p* = 0.35) of sensitivity, and neither self-compassion nor depressive symptoms were statistically significant predictors of sensitive parenting after adjustment for covariates. Considerations for future studies are discussed.

## 1. Introduction

Parenting is a complex phenomenon that involves a multifaceted set of behaviors influenced by numerous intersecting factors and environments [1]. There is consistent evidence that the ways in which parents interact with children are related to children’s adjustment in multiple domains including the development and maintenance of psychopathology [2]. Early parenting research focused on direct means of socialization through parenting styles and practices that describe characteristics of the interactions between parent and child. Initially described by Baumrind (1966) and expanded upon by Maccoby and Martin (1983), this foundational work provides a framework for organizing parenting styles along two orthogonal dimensions of demandingness and responsiveness [3,4,5].

While cultural differences are related to parenting practices [6], a substantial body of research using an attachment perspective has consistently shown that parenting characterized by the caregiver’s accessibility, acceptance, cooperation, and responsivity to children’s bids for attention and care as optimal for child development [7]. Reviews of the child development literature point to a parent’s ability to perceive and read their children’s signals, interpret them adequately and respond promptly, called parental sensitivity, as an essential aspect of parenting that impacts children’s socioemotional development [8,9]. As evidence mounts on the key role of parenting behavior to child health and development, a task for researchers is to determine the factors related to sensitive, responsive caregiving. Belsky’s (1984) influential theoretical framework for understanding determinants of parenting behavior and its recent update [1,10] converge on three broad categories of key predictors of parenting: characteristics of the parent (e.g., depression and depressive symptoms, education), characteristics of the child (e.g., child temperament, sex, age), and contextual sources of support and stress (e.g., social networks). Key contextual factors in the family’s ecological network, including family poverty, neighborhood characteristics, and ethnicity/culture, were not included in Belsky’s initial model but have been added in subsequent work [10].

In particular, Belsky proposed that caregiver characteristics (e.g., maternal mental health, level of education) would be the most influential in predicting parenting behavior. Several decades of research supports this proposition reporting robust links between caregiver characteristics, such as coping styles and depressive symptoms, are related to parenting such that ineffective coping styles [11,12] and depressive symptoms [13] predict lower-quality parenting. Additional research shows that sociodemographic factors, such as education and income, are positively associated with sensitive parenting such that higher levels of education and income are related to greater sensitivity. Since its inception, Belsky’s model has been applied and expanded upon in various contexts [10,14]. Although the literature on mindfulness and parenting was presented almost two decades ago, the role of self-compassion on parenting, however, is very unclear [15].

### 1.1. Mindful Self-Compassion

Self-compassion is defined as a way of treating oneself with love, support, and care when facing challenges and stress [16,17,18]. Although a relatively new construct in psychological research, there is growing interest in self-compassion. Rooted in Buddhist philosophy, Neff conceptualized self-compassion as a coping and emotional regulation strategy used by people when they are experiencing hardship, failures, feelings of inadequacy, or other general life challenges [17,19].

Much of the literature on self-compassion examines the benefits of self-compassion on individual well-being with one of the most consistent findings in the literature being the inverse relation between self-compassion and anxiety and depression [20]. This body of research suggests that self-compassion is an adaptive emotion regulation strategy that may exert a protective effect in the presence of vulnerability factors such as depressive symptoms and anxiety [21,22,23]. This line of thinking posits that in contrast to individuals low on SC, those with high levels of self-compassion might view negative experiences as more controllable and less aversive and thus potentially provides a buffer against the activation of schema related to psychopathology [22,23]. In addition, there is growing evidence suggesting that self-compassion is linked to interpersonal relationships and therefore may be an important characteristic to consider as a determinant of parenting sensitivity.

Three components make up Neff’s definition of self-compassion: self-kindness, shared humanity, and mindfulness. Self-kindness is described as extending warm soothing responses to oneself as compared to self-criticism. A second component is common humanity defined as viewing one’s difficult experiences as part of the larger human experience and recognizing that all people encounter hardships and emotional distress in their lives. The third component is mindfulness, which refers to a balanced and non-judgmental response to negative emotions, as opposed to avoiding or becoming overwhelmed by them [17].

### 1.2. Self-Compassion, Parenting, Depressive Symptoms and SES

Given the basic tenets of self-compassion include a healthy attitude towards oneself during times of distress or challenge, numerous studies examined self-compassion in relation to a range of factors that predict parenting quality. Two studies using cross-sectional designs reported that mothers’ adult attachment style was associated with their self-compassion and parenting behavior such that insecure attachment was inversely related to self-compassion [24,25]. Additional research suggests that depressed parents are more self-critical and less positive towards their children [26,27]. Further, Felder et al., using a sample of postpartum women, reported that the severity of depression was related to self-compassion such that greater depressive symptomatology was related to lower self-compassion [28].

Similarly, Gouveia and colleagues found that parents high in self-compassion report lower levels of authoritarian parenting (involving strict order and rule enforcement and low warmth and support) and permissive parenting (characterized by excessive tolerance, indulgence and absence of rules or punishment) [29,30]. Only one study of depressed parents (*n* = 36 mothers, and 2 fathers) used observational assessments of parents interacting with their children ages 2–6 and reported that self-compassion was not related to sensitive parenting [26]. Much of the current research on self-compassion and parenting has focused on new mothers in the postpartum phase or with older children in their youth, and very few studies have examined the relations between self-compassion and parenting in the toddler phase.

In light of the theory and prior work highlighted above, the present study addresses an important gap in the literature by examining the associations between maternal sensitivity, maternal depression, sociodemographic variables and self-reported maternal self-compassion. We use Belsky’s conceptualizations of determinants of parenting behavior that proposed that characteristics of the caregiver were the most influential for parental functioning. Substantial evidence shows that parental depression is associated with deficits in parenting, including decreased warm and responsive parenting behaviors [31,32]. Both self-report and observational studies have demonstrated that the behavior of depressed parents is characterized by diminished warmth and responsiveness, and higher rates of hostility in interactions with their children throughout development [33]. Similarly, family stress models show that sociodemographic factors have a strong influence on parenting [34,35,36]. For example, Black, Indigenous, and other minority populations experience greater stress in the parenting role given historical structural disparities in how societies have supported families [37,38]. However, no study, of which we are aware, has considered whether sociodemographic characteristics of the caregiver, such as education or household income, are related to self-compassion in a demographically diverse sample of mothers. Based on previous research, we examined associations between self-compassion, depressive symptoms and sensitive parenting, hypothesizing that mothers with high levels of self-compassion would have lower ratings of depressive symptoms and would be observed to be more sensitive in their parenting behavior while playing with their toddler. Given previous reports linking self-compassion and depressive symptoms to parenting behavior, we examined the direct and interactive effects of self-compassion and depressive symptoms on observed parenting behavior [39,40].

## 2. Materials and Methods

### 2.1. Methods

#### Sample and Procedure

Data for this study are from the toddler-phase of an observational cohort of 300 parent–child dyads recruited from a major pediatric care system in Central Ohio between December 2017 and May 2019. Children were between 16–20 months of age at time of recruitment. Inclusion criteria specified that families lived within 15 miles of the main campus of Nationwide Children’s Hospital (NCH; Columbus, OH, USA), the primary caregiver spoke English, and that the caregiver was present for mealtimes with the child at least a few times a week. The cohort includes children of all gestational ages with an over-representation of children who were born preterm (<37 weeks’ completed gestation). The study protocol was approved in March 2017 by the Nationwide Children’s Hospital Institutional Review Board (IRB16-00826), and all caregivers provided written informed consent. Additional details on cohort recruitment and study visit protocols have been published [41].

The data utilized in our analysis were collected during two toddler-phase study visits. The first of these coincided with enrollment into the cohort (when the child was an average calendar age of 18 months) and took place in a behavioral laboratory setting. The second visit occurred in the participant’s home and was scheduled for plus/minus two weeks of the child’s 2nd birthday. Each visit involved questionnaires, observations and assessments and lasted approximately 90 min. With the exception of maternal self-compassion, which was assessed as a component of the second visit, data used in this analysis are from the first study visit. Our analysis should be treated as cross-sectional because in the absence of significant life events, self-compassion is stable over time [17,42].

### 2.2. Measures

Maternal depressive symptoms. At the 18-month laboratory visit, the primary caregiver completed the 20-item Center for Epidemiologic Studies-Depression Scale (CES-D) [43] to measure depressive symptomology over the previous week. Respondents indicated how often they have felt or behaved in certain ways ranging from “1 = rarely or none of the time (less than 1 day)” to “4 = most of all of the time (5–7 days)”. A sample item is “I felt that I could not shake off the blues even with the help of my family or friends”. Positively worded items are reversed scored, and higher scores represent more depressive symptoms overall. Previous studies have reported good reliability and validity for this measure, as well as high internal consistency.

Maternal sensitive parenting behavior. At the conclusion of the 18-month laboratory visit, the primary caregiver and toddler participated in a standardized 10 min free-play activity, which was digitally recorded. The mother–child dyad was seated on a mat, and a research assistant presented them with a developmentally appropriate set of toys. Mothers were instructed to interact with the child as they typically would if given some free time during the day. Staff monitored the room remotely during the 10 min interaction. Digital video recordings were securely transferred for behavioral coding by trained staff supervised by an expert coder. Using four global rating scales (sensitivity/responsiveness, detachment/disengagement, stimulation of cognitive development, and positive regard) adapted from those used by the National Institute for Child Health and Human Development Study of Early Child Care (NICHD ECCRN, 1999), coders rated parenting behaviors on a seven-point scale (on which 1 = not at all characteristic and 7 = very characteristic) [44,45]. Both frequency and intensity of behavior or affect toward the child were considered. Inter-rater reliability was assessed using Intraclass Correlations (ICCs). Coders underwent training until acceptable reliability (ICC > 0.70) was achieved and maintained for each coder on every scale. Once acceptable reliability was established, coders began coding in pairs while continuing to code at least 20% of their weekly cases with a criterion coder. Each coding pair met biweekly to reconcile scoring discrepancies; the final scores that they arrived at by consensus were used in all analyses.

The sensitivity/responsiveness scale, adapted from Ainsworth et al., describes maternal behavior characterized by an awareness and responsive to the child’s cues and bids for attention and care [46]. The detachment/disengagement scale describes the degree to which the parent was emotionally distant, disengaged, or unaware of the child’s signals or needs for appropriate facilitation or care. The stimulation of cognitive development scale measures the degree to which the parent engaged in age-appropriate behaviors that foster cognitive and physical development of the child. The positive regard scale rates the quantity and intensity of the parent’s expression of positive feelings toward the child, including praise, smiling, physical affection, playful behavior and overall enjoyment, autonomy and self-awareness.

Sensitive parenting consisted of the mean values of the reverse score for the detachment/disengagement scale and the scores for sensitivity/responsiveness, positive regard and stimulation of cognitive development scales. Accordingly, higher scores on the sensitivity subscale reflect parenting behaviors that are child-centered, engaged, warm and stimulating. These measures have been successfully tested in other investigations [32,47]. Inter-rater reliability was monitored throughout the coding period with intraclass correlations ranging from 0.79–0.90.

Maternal self-compassion. Caregivers were invited to complete self-report questionnaires online before the 24-month home visit, including the Self-Compassion Scale-Short Form which was modified to lower the reading comprehension level [48,49]. This self-report measure consists of 12 items, each rated on a five-point Likert scale (1 = almost never to 5 = almost always), with questions such as “When something upsets me I try to keep my emotions in balance” and “When I feel that I have done poorly, I try to remind myself that many people also feel this way”. An overall summative index of self-compassion is determined after reverse coding six items (e.g., “I get frustrated by those parts of my personality I don’t like”). Higher scores represent greater self-compassion. The Self-Compassion Scale is a widely used, valid and reliable measure, and displayed high internal consistency in the present sample (Cronbach’s α = 0.82).

Sociodemographic characteristics. Children’s date of birth, sex, and gestational age were abstracted from the medical record and confirmed with the caregiver at the first study visit. The caregiver reported additional demographic characteristics of the family and household during the first study visit.

### 2.3. Analytic Approach

Our analytic sample included 274 mother–toddler dyads. We excluded from analyses the small number of dyads (*n* = 17) in which the caregiver was not the child’s mother or for which we did not have complete data for all variables used in analyses (*n* = 9). We examined the distribution of depressive symptoms (CES-D score), maternal sensitivity and self-compassion and the correlations between these variables. We describe how maternal self-compassion is related to sociodemographic characteristics of the sample including maternal and child age at the first study visit, child sex, gestational age, child race/ethnicity, maternal education level and annual household income. To aid interpretability, we categorized self-compassion as low, average or high based on whether self-compassion scores were below, within or above one standard deviation of the sample mean. Analysis of variance and chi-square tests were used to test for differences in self-compassion across groups defined by sociodemographic characteristics. To avoid small cell sizes in these analyses, we collapsed some levels of sociodemographic characteristics. Linear regression models with parental sensitivity as the outcome variable were estimated to assess self-compassion and depressive symptoms as predictors. Each was modeled separately, together and in interaction, with and without adjustment for sociodemographic characteristics (maternal age, education, child age, race and household income). Prior research indicates associations between these factors and parenting quality [35,50]. Analyses were conducted in SAS with an alpha level of 0.05.

## 3. Results

Characteristics of the sample are shown in Table 1. The mean age of mothers was 30.6 (SD = 5.8) years. The cohort is diverse relative to race and ethnicity, maternal educational attainment and household income (Table 1). Children (57% male) ranged in calendar age from 16.5 to 20.0 months with a mean (SD) of 18.2 (0.7) months. Thirty-eight percent were born preterm (<37 completed weeks’ gestation). Self-Compassion Scale scores ranged from 17 to 60 with a mean (SD) of 42.5 (7.0). The mean (SD) level of maternal depressive symptoms (CES-D) was 7.7 (6.3) and ranged from 0 to 37 (Table 1).

Self-compassion was strongly related to depressive symptoms (*r* = −0.42, *p* < 0.0001) such that mothers who reported higher levels of depressive symptoms reported lower levels of self-compassion. Depressive symptoms were also associated with lower levels of parental sensitivity (r = −0.12, *p* = 0.046). However, self-compassion was not related to sensitive parenting (r = 0.00, *p* = 0.99).

### 3.1. Associations between Self-Compassion and Sociodemographic Factors

We examined sociodemographic factors across levels of self-compassion and defined as below 1 SD of the mean (range 17–35, *n* = 50), within 1 SD of the mean (range = 36–49, *n* = 181) and above 1 SD of the mean (range = 50–60, *n* = 43). Table 2 describes sample characteristics within each of the three levels of self-compassion. There was no evidence that self-compassion was associated with any of the sociodemographic characteristics we examined (Table 2). The distribution of sociodemographic characteristics was similar among those mothers with low, average or high levels of self-compassion (Table 2). For example, among mothers with low self-compassion levels, 30% had annual household income below $20,000 and 26% had annual household income of $90,000 or more. These percentages were 24% and 23% in the average self-compassion group, and 35% and 26% in the high self-compassion group (Table 2). As expected, the mean level of maternal depressive symptoms decreased across increasing levels of self-compassion (*p* < 0.0001).

### 3.2. Self-Compassion and Depressive Symptoms as Predictors of Sensitive Parenting

Estimates from linear regression models predicting parenting sensitivity are shown in Table 3. Modeling self-compassion and depressive symptoms individually with and without adjustment for sociodemographic covariates (Univariate Models, Table 3), we observed that greater depressive symptoms were associated with lower parenting sensitivity in the unadjusted model (estimate = −0.029, SE = 0.015, *p* = 0.046), but this association was attenuated and not statistically significant after adjustment for sociodemographic characteristics (B = −0.018, SE = 0.012, *p* = 0.14). Self-compassion was not related to sensitive parenting in unadjusted or adjusted models. When self-compassion and depressive symptoms were both included as predictors of parenting sensitivity, depressive symptoms were related to sensitive parenting (B = −0.036, SE = 0.016, *p* = 0.03), but self-compassion was not a statistically significant predictor (*p* = 0.35). However, consistent with the univariate models, neither self-compassion nor depressive symptoms were statistically significant predictors after adjustment for covariates. We then examined whether self-compassion and depressive symptoms interacted in the prediction of sensitive parenting. We found no evidence of an interaction between self-compassion and depressive symptoms in the prediction of sensitive parenting (p interaction = 0.73).

## 4. Discussion

A substantial literature documents that the ways in which parents interact with their children, i.e., the quality of parent–child interactions, is related to optimal child development. According to Belsky, multiple factors affect parenting and child outcomes, with parental psychological well-being playing a prominent role [1]. Indeed, empirical research has supported parental psychological distress (e.g., depression) as a parental risk factor for maladaptive fathering and mothering [33,40,51]. The present study adds to the growing literature on the associations between self-compassion and parenting behavior. Using observational assessments of sensitive parenting and self-report on self-compassion and depressive symptoms along with socio-demographic factors, the findings from this study indicate that although self-compassion is related to depressive symptoms, it was not related to observed sensitive parenting behavior or socio-demographic factors.

Belsky’s model posits that parents’ personal psychological resources are the most influential determinants of parenting behavior. Our findings differ from those of previous studies proposing an association between self-compassion and sensitive parenting behavior. One possible explanation for the differences in the findings from these earlier studies and the present study may be due methodological constraints given that much of the current evidence linking self-compassion to parenting behavior are based solely on mothers’ reports of their own characteristics (i.e., parenting). The issue of rater bias, or common method variance, may have influenced these previous results [52].

With regard to self-compassion and depressive symptoms, our results suggest that mothers who practice self-compassion, on average, tend to report lower levels of depressive symptoms such as negative affect. In keeping with previous literature, this finding was replicated in our analyses. The null findings that self-compassion did not predict parenting or depressive symptoms nor the lack of interactive effect may be reflective of differences in the characteristics of the samples included. Much of the available evidence comes from small studies of homogeneous populations of highly educated individuals with above average incomes or from clinical studies. The current study, however, was conducted in a large diverse sample.

In contrast to previous research linking self-compassion and sensitive, responsive parenting behavior, we found no significant association between the two. One possible explanation for the discordant finding is that much of the research on self-compassion and parenting has used self-report assessments from parents and potential response biases may have occurred when answering sensitive questions. In the current study, we used observational assessments of parents and their toddlers interacting with each other. Observational techniques provide a window on actual behaviors of interest (e.g., warmth, responsiveness and attentiveness). In addition, although there are other dimensions of parenting (i.e., strictness and demandingness), in the current study, we focused on one dimension of parenting, i.e., parental sensitivity. It may be that self-compassion may have differential effects on other dimensions of parenting. Furthermore, although our coding paradigm is designed to capture a broad range of parenting behaviors in the context of mildly challenging tasks (e.g., puzzles and building blocks), they are not designed to elicit distress per se. It may be that the associations between sensitive, responsive caregiving and self-compassion may be more salient in the context of child distress as it can help mothers regulate their own emotions and respond to their child with empathy and compassion [16]. Future research may benefit from observing sensitivity in mothers in interactions with young children in frustrating tasks that may evoke greater emotional response to child needs [53].

Given that much of the existing evidence linking self-compassion to parenting has been with older children and adolescents, it might also be the case that self-compassion may be more important in these developmental stages. Although previous studies reported that sociodemographic factors such as income and education may be related to self-compassion in mothers such that greater income and education are associated with greater self-compassion, the current study does not support these previous findings. Although we did note a trend (*p* = 0.07) for maternal age, we did not find that education or income were related to self-compassion across levels of SC.

While this study has numerous strengths including the use of a large, diverse sample, and observational assessments of sensitive parenting behavior, limitations must also be noted. In our study, observed parenting occurred at the 18-month (child age) study time point whereas parents completed the self-compassion assessment on the 24-month visit. Based on work by Medvedev and colleagues, we posit that self-compassion is a trait rather than a state, making it time-stable and that measurement at 24 months is equivalent to what it would have been had we assessed it at 18 months [54]. This perspective is supported by test–retest reliability data produced by Neff in the original development of the scale and demonstrates the SCS-SF as a valid and reliable measure of trait characteristics of self-compassion [16]. Future studies may benefit from using a multi-method, multi-informant approach to understand the relationship between self-compassion and parenting behavior given the bias stemming from a single-informant approach. Further, our sample was comprised of participants from a large Midwestern US city, and the results may not be generalizable to the larger population in the US or to a wider population or different cultural settings. Moreover, this study did not explore the associations between the three dimensions of self-compassion (self-kindness versus self-judgment, common humanity versus isolation and mindfulness versus over-identification) and the other study variables. Future studies should explore the specific role of each dimension of self-compassion as it may relate to additional domains of parenting, which will allow a more comprehensive understanding of the associations between self-compassion and parenting outcomes that, in turn, could help practitioners understand how best to tailor interventions.

## Figures and Tables

**Table 1 children-10-01284-t001:** Participant characteristics at first study visit at child age of approximately 18 months (analysis sample, *n* = 274).

Characteristic	Mean (SD)	Range	N (%)
Age (years) of mother	30.6 (5.8)	18–47	
Age (months) of child	18.2 (0.7)	16.5–20.0	
Child Sex			
Boy			156 (57%)
Girl			118 (43%)
Gestational age (completed weeks)	36.3 (4.4)	23–41	
Preterm (<37 weeks)			
(0) No			171 (62%)
(1) Yes			103 (38%)
Child race (NH White)	121 (44%)		
(2) NH Black	99 (36%)		
(3) Other + Hispanic	54 (20%)		
Mother’s education level			
(1) Less than high school	13 (5%)		
(2) High school diploma/GED	49 (18%)		
(3) Some college but not AA	72 (26%)		
(4) Associates degree	25 (9%)		
(5) Bachelor’s degree	58 (21%)		
Household income, collapsed			
(1) <20 K	73 (27%)		
(2) 20 to <40 K	58 (21%)		
(3) 40 to <60 K	37 (14%)		
(4) 60 K to <90 K	41 (15%)		
(5) 90 K+	65 (24%)		
Measures			
CES-D score (T1)	7.7 (6.3)	0–37	
PCI: Sensitive Composite *	3.9 (1.5)	1–7	
Self-Compassion *	42.7 (6.93)	17–60	

* Note: Assessed at child age of 24 months.

**Table 2 children-10-01284-t002:** Comparison of demographics and key study variables across Self-Compassion Scale (SCS) categories.

	Categorized Self-Compassion Scale			
	Low (n = 50)	Average SCS (n = 181) Mean (SD)	High SCS (n = 43) Mean (SD)	
Characteristic	or N (%)	or N (%)	or N (%)	p-Value *
Demographics				
Age (years) of mother (T1)	29.7 (6.6)	31.2 (5.5)	29.3 (6.0)	0.07
Age (months) of child (T1)	18.3 (0.7)	18.2 (0.7)	18.1 (0.7)	0.30
Gestational age (completed weeks)	37.2 (3.4)	36.1 (4.5)	36.2 (4.7)	0.30
Preterm (<37 weeks)				0.74
(0) No	33 (66%)	110 (61%)	28 (65%)	
(1) Yes	17 (34%)	71 (39%)	15 (35%)	
Child gender				0.86
(1) Boy	30 (60%)	101 (56%)	25 (58%)	
(2) Girl	20 (40%)	80 (44%)	18 (42%)	
Child race				0.56
(1) NH White	25 (50%)	81 (45%)	15 (35%)	
(2) NH Black	18 (36%)	64 (35%)	17 (40%)	
(3) Other + Hispanic	7 (14%)	36 (20%)	11 (26%)	
Mother’s education level, collapsed				0.89
(1) HS or less	10 (20%)	40 (22%)	12 (28%)	
(2) Some college/Assoc deg	19 (38%)	63 (35%)	15 (35%)	
(3) Bachelors+	21 (42%)	78 (43%)	16 (37%)	
Household income, collapsed				0.39
(1) <20 K	15 (30%)	43 (24%)	15 (35%)	
(2) 20 to <50 K	10 (20%)	62 (34%)	11 (26%)	
(3) 50 to <90 K	12 (24%)	35 (19%)	6 (14%)	
(4) 90 K+	13 (26%)	41 (23%)	11 (26%)	

* From chi-square test or ANOVA comparing across the three SCS categories.

**Table 3 children-10-01284-t003:** Estimated slopes using CES-D and Self-Compassion Scale (SCS) to predict parental sensitivity.

	Unadjusted			Adjusted *		
	Estimate	(SE)	*p*-Value	Estimate	(SE)	*p*-Value
Model 1: Each alone						
SCS	−0.00008	(0.013)	0.995	0.0097	(0.011)	0.36
CES-D	−0.029	(0.015)	0.046	−0.018	(0.012)	0.14
Model 2: Both CES-D and SCS						
CES-D	−0.036	(0.016)	0.03	−0.016	(0.013)	0.23
SCS	−0.014	(0.015)	0.35	0.0035	(0.012)	0.76
Model 3: Interaction						
CES-D **	−0.033	(0.018)	0.06	−0.011	(0.015)	0.46
SCS **	−0.013	(0.015)	0.36	0.0043	(0.012)	0.72
CES-D × SCS	0.00078	(0.0023)	0.73	0.0015	(0.0018)	0.42

* Adjusted for parental education (three levels), household income (five levels) and child’s race (three levels). ** Centered at sample mean for ease of interpretation.

## Data Availability

Data are available upon reasonable request. The datasets generated and/or analysed during the current study are not publicly available unless approved by the Institutional Review Board overseeing this study but may be available from the corresponding author on reasonable request.

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
