# Peer review of "Self-Compassion and Depressive Symptoms as Determinants of Sensitive Parenting: Associations with Sociodemographic Characteristics in a Sample of Mothers and Toddlers"

_children, 2023, doi:10.3390/children10081284_

Round 1

Reviewer 1 Report

This is a very strong manuscript that makes an important contribution to the fields of parenting and child well-being. It is well-written and clear.

The literature review is very good, utilizing 51 references. The reliance upon Belsky's model is certainly appropriate for this work. And examining self-compassion on the part of parents offers to extend knowledge on successful parenting.

The section on Mindful Self-Compassion is especially strong. The authors nicely define self-compassion and its component parts: self-kindness, shared humanity, and mindfulness.

Line #95 should be "Given that the"...

The sample of 300 parent-dyads is very good with 274 mother-toddler triads. The authors later make the point that this study is distinguished from others by relying upon observations of interactions and not self-reports from parents. The authors note that the large and diverse sample is a strength of their work.

The Measures section is clear, with the use of appropriate scales: the Center for Epidemiological Studies-Depression Scale, the adaptation of the 4 global rating scales from the National Institute for Child Health and Human Development Study of Early Child Care, the adapted sensitivity and responsiveness scale based on the work of Ainsworth et al., and the Self-Compassion Scale-Short form.

Line #202: "overall enjoyment, autonomy"...

Line #280: "unadjusted or unadjusted models" needs correction.

The Results section is strong. I am not a research methodologist/statistician and I hope another reviewer can be helpful here.

The Discussion section is very strong. We learn that depressive symptoms are related to self-compassion but that observed parenting behaviors and socio-demographic factors are not related to self-compassion. The authors are clear that some of their research findings do not concur with previous research--self-compassion was not related to sensitive parenting.

Line # 346: "exiting" should be "existing"

The end of the article is strong--the authors note the limitations of their work and they offered important recommendations for how to extend their work. Overall, very good work!

Author Response

We thank the reviewer for their careful reading of the manuscript and have made the suggested edits. In addition:

  1. Regarding Line #202: "overall enjoyment, autonomy"; We have edited this sentence to end with ..overall enjoyment.
  2. We believe this makes it more clear to the reader and thank the reviewer for their thoughtful reading of the manuscript.

Reviewer 2 Report

Dear Authors

Very interesting article with much vulnerabilities that need revision. For example,

The title has no clear exposition and outcome.

Why did you put the title again before the introduction?

References are misplaced in the text. They are not placed after the fullstop. See the author guidelines.

the «current study» section is not needed.

The purpose of your report is not clear.

I also didn't understand the exclusion criteria. Can you explain the reasons you excluded allergy problems and pregnancies >42 weeks?

Why are measures being developed so much?

Table 1 could be omitted as long as the data are reported in the text.

The tables are also confusing as they are distributed. It is good that each table with its commentary goes together.

The discussion and conclusions are little commented. I think the article needs a complete revision

Good luck!

 Minor editing of English language required

Author Response

Reviewer 2:

We thank the reviewer for their careful reading of the manuscript and have made the suggested edits.

  1. We removed the “Current Study” heading.
  2. References have been edited to journal specifications.
  3. We have streamlined the methods section relating to exclusion criteria to address reviewer concerns.
  4. We included the title before the introduction due to APA guidelines. We will seek guidance from journal editors and make corrections as required.

Reviewer 3 Report

Manuscript Number 2440284

Title: “Self-compassion and depressive symptoms as determinants of 2 parenting: Associations with sociodemographic characteristics 3 in a sample of mothers and toddlers”

The present manuscrits aims to examine the associations between self-compassion, depressive symptoms, socioeconomic status, and sensitive parenting. Participants were obtained by a cohort study of 300 families in central Ohio enrolled when children were a mean (SD) calendar age of 18.2 (0.7) months. Results showed that depressive symptoms were related to sensitive parenting.

Theoretical part

The present manuscript discusses parenting and mentions some of the parenting styles in the introduction without referring to the model that has had the greatest impact on the literature on parenting styles, the two-dimensional theoretical model of Maccoby and Martin. Therefore, the manuscript requires further theoretical conceptualization about the two main parental dimensions, the four parental styles and the theoretical debate about which parental style is the best in different cultural contexts.

There are other theoretical models of parenting that have not been mentioned, such as Baumrind's Y model, which should not be ignored.

One of the firsts models about parenting was Baumrind´s Y model (Baumrind, 1968). This model edfends the role of parental control. She proposed three parental styles: authoritative, authoritarian, and permissive; which corresponded to three modes of parental control, the authoritative control, the authoritarian control, and the lack of control (i.e., permissive control) (Baumrind, 1968).

Maccoby and Martin's two-dimensional (Maccoby & Martin, 1983) was the model that has had the greatest impact on the parental socialization literature and became the reference model in the parenting literature. This model state that there two main independent parenting dimensions that are used by parents to socialize their children: (i.e., warmth and strictness) (Alcaide et al., 2023; Climent-Galarza et al., 2022; Darling & Steinberg, 1993; Fuentes et al., 2022; F. Garcia & Gracia, 2009; Gimenez-Serrano et al., 2022; Lamborn et al., 1991; Martinez et al., 2020; Martinez-Escudero et al., 2023; Martínez et al., 2021; Palacios et al., 2022; Queiroz et al., 2020).

On the one hand, parental warmth refers to that parents show acceptance,love, affection, emotional closeness, and communication with their children (Alcaide et al., 2023; Climent-Galarza et al., 2022; Lamborn et al., 1991; Martinez et al., 2020; Martinez-Escudero et al., 2020; Martínez et al., 2021). Other labels used to warmth are responsiveness, involvement, implication, security, care or love (Darling & Steinberg, 1993; F. Garcia & Gracia, 2014; Martinez et al., 2020). On the other hand, parental strictness refers to the extent of imposition, authority, or rigidity (Climent-Galarza et al., 2022), the use discipline towards their children, controlling and/or supervising their behavior, establishing norms for children’s behavior, and maintaining position of authority (Alcaide et al., 2023; Darling & Steinberg, 1993).

Maccoby and Martin's two-dimensional model also defends the idea that four parenting styles emerges from the combination of the two main parenting dimensions (i.e., warmth and strictness): authoritarian (strictness but not warmth); authoritative (strictness and warmth), indulgent (warmth but not strictness) and neglectful (neither strictness nor warmth) (Alcaide et al., 2023; Climent-Galarza et al., 2022; Fuentes et al., 2022; O. F. Garcia et al., 2020; Gimenez-Serrano et al., 2022; Maccoby & Martin, 1983; Martinez-Escudero et al., 2023; Palacios et al., 2022; Perez-Gramaje et al., 2020; Queiroz et al., 2020; Villarejo et al., 2020).

The two-dimensional model (Maccoby & Martin, 1983) gave rise to a theoretical debate about which parenting style is best.

Classical studies about parenting conducted in Anglo-Saxon contexts with European-American samples (mostly white middle-class families) stated that the authoritative parenting (i.e., parental warmth and parental strictness together) was related to the best child psychosocial adjustment (Baumrind, 1991; Lamborn et al., 1991; Steinberg et al., 1991; Steinberg et al., 1994). These results were not always the same in all cultural contexts. Other studies conducted in ethnic minority groups in the United States such as Chinese Americans (Chao, 2001) or African American (Deater-Deckard et al., 1996), and Arabs societies (Dwairy & Achoui, 2006) defended that the authoritarian parenting (i.e., parental strictness without parental warmth) is related to the best child adjustment. Contrary to classical studies, the most recent evidence conducted in European and Latin American countries support the idea that indulgent parenting (i.e, parental warmth without parental strictness) is related to the best child psychosocial adjustment (Alcaide et al., 2023; Fuentes et al., 2022; F. Garcia & Gracia, 2009; O. F. Garcia et al., 2018; O. F. Garcia & Serra, 2019; O. F. Garcia et al., 2020; Gimenez-Serrano et al., 2022; Gimenez-Serrano, Garcia et al., 2022; Martinez-Escudero et al., 2020; Martinez-Escudero et al., 2023; Martínez et al., 2019; Palacios et al., 2022; Perez-Gramaje et al., 2020; Villarejo et al., 2020).

Empirical part

The structure of the method section is in accordance with that usually used in scientific articles.

On the measure "maternal self-compassion", it could be specified whether high scores on the questionnaire are related to high or low scores on the measure "maternal self-compassion".

References

Alcaide, M., Garcia, O. F., Queiroz, P., & Garcia, F. (2023). Adjustment and maladjustment to later life: Evidence about early experiences in the family. Frontiers in Psychology, 14https://doi.org/10.3389/fpsyg.2023.1059458

Baumrind, D. (1968, Authoritarian vs authoritative parental control. Adolescence, 3, 255-272.

Baumrind, D. (1991). Effective parenting during the early adolescent transition. In P. A. Cowan, & E. M. Herington (Eds.), Advances in family research series. Family transitions (pp. 111-163). Lawrence Erlbaum Associates, Inc.

Chao, R. K. (2001). Extending research on the consequences of parenting style for Chinese Americans and European Americans. Child Development, 72, 1832-1843. https://doi.org/10.1111/1467-8624.00381

Climent-Galarza, S., Alcaide, M., Garcia, O. F., Chen, F., & Garcia, F. (2022). Parental socialization, delinquency during adolescence and adjustment in adolescents and adult children. Behavioral Sciences, 12(11)https://doi.org/10.3390/bs12110448

Darling, N., & Steinberg, L. (1993). Parenting style as context: An integrative model. Psychological Bulletin, 113(3), 487-496. https://doi.org/10.1037/0033-2909.113.3.487

Deater-Deckard, K., Dodge, K. A., Bates, J. E., & Pettit, G. S. (1996). Physical discipline among African American and European American mothers: Links to children's externalizing behaviors. Developmental Psychology, 32(6), 1065-1072. https://doi.org/10.1037/0012-1649.32.6.1065

Dwairy, M., & Achoui, M. (2006). Introduction to three cross-regional research studies on parenting styles, individuation, and mental health in Arab societies. Journal of Cross-Cultural Psychology, 37, 221-229. https://doi.org/10.1177/0022022106286921

Fuentes, M. C., Garcia, O. F., Alcaide, M., Garcia-Ros, R., & Garcia, F. (2022). Analyzing when parental warmth but without parental strictness leads to more adolescent empathy and self-concept: Evidence from Spanish homes. Frontiers in Psychology, 13https://doi.org/10.3389/fpsyg.2022.1060821

Garcia, F., & Gracia, E. (2014). The indulgent parenting style and developmental outcomes in South European and Latin American countries. In H. Selin (Ed.), Parenting Across Cultures (pp. 419-433). Springer. https://doi.org/10.1007/978-94-007-7503-9_31

Garcia, F., & Gracia, E. (2009). Is always authoritative the optimum parenting style? Evidence from Spanish families. Adolescence, 44(173), 101-131.

Garcia, O. F., Fuentes, M. C., Gracia, E., Serra, E., & Garcia, F. (2020). Parenting warmth and strictness across three generations: Parenting styles and psychosocial adjustment. International Journal of Environmental Research and Public Health, 17(20), 7487. https://doi.org/10.3390/ijerph17207487

Garcia, O. F., & Serra, E. (2019). Raising children with poor school performance: Parenting styles and short- and long-term consequences for adolescent and adult development. International Journal of Environmental Research and Public Health, 16(7), 1089. https://doi.org/10.3390/ijerph16071089

Garcia, O. F., Serra, E., Zacares, J. J., & Garcia, F. (2018). Parenting styles and short- and long-term socialization outcomes: A study among Spanish adolescents and older adults. Psychosocial Intervention, 27(3), 153-161. https://doi.org/10.5093/pi2018a21

Gimenez-Serrano, S., Alcaide, M., Reyes, M., Zacarés, J. J., & Celdrán, M. (2022). Beyond parenting socialization years: The relationship between parenting dimensions and grandparenting functioning. International Journal of Environmental Research and Public Health, 19(8), 4528. https://doi.org/10.3390/ijerph19084528

Gimenez-Serrano, S., Garcia, F., & Garcia, O. F. (2022). Parenting styles and its relations with personal and social adjustment beyond adolescence: Is the current evidence enough? European Journal of Developmental Psychology, 19(5), 749-769. https://doi.org/10.1080/17405629.2021.1952863

Lamborn, S. D., Mounts, N. S., Steinberg, L., & Dornbusch, S. M. (1991). Patterns of competence and adjustment among adolescents from authoritative, authoritarian, indulgent, and neglectful families. Child Development, 62(5), 1049-1065. https://doi.org/10.1111/j.1467-8624.1991.tb01588.x

Maccoby, E. E., & Martin, J. A. (1983). Socialization in the context of the family: Parent–child interaction. In P. H. Mussen (Ed.), Handbook of child psychology (pp. 1-101). Wiley.

Martínez, I., Murgui, S., Garcia, O. F., & Garcia, F. (2021). Parenting and adolescent adjustment: The mediational role of family self-esteem. Journal of Child and Family Studies, 30(5), 1184-1197. https://doi.org/10.1007/s10826-021-01937-z

Martínez, I., Murgui, S., Garcia, O. F., & Garcia, F. (2019). Parenting in the digital era: Protective and risk parenting styles for traditional bullying and cyberbullying victimization. Computers in Human Behavior, 90, 84-92. https://doi.org/10.1016/j.chb.2018.08.036

Martinez, I., Garcia, F., Veiga, F., Garcia, O. F., Rodrigues, Y., & Serra, E. (2020). Parenting styles, internalization of values and self-esteem: A cross-cultural study in Spain, Portugal and Brazil. International Journal of Environmental Research and Public Health, 17(7), 2370. https://doi.org/10.3390/ijerph17072370

Martinez-Escudero, J. A., Garcia, O. F., Alcaide, M., Bochons, I., & Garcia, F. (2023). Parental socialization and adjustment components in adolescents and middle-aged adults: How are they related? Psychology Research and Behavior Management, 16, 1127-1139. https://doi.org/10.2147/PRBM.S394557

Martinez-Escudero, J. A., Villarejo, S., Garcia, O. F., & Garcia, F. (2020). Parental socialization and its impact across the lifespan. Behavioral Sciences, 10(6), 101. https://doi.org/10.3390/bs10060101

Palacios, I., Garcia, O. F., Alcaide, M., & Garcia, F. (2022). Positive parenting style and positive health beyond the authoritative: Self, universalism values, and protection against emotional vulnerability from Spanish adolescents and adult children. Frontiers in Psychology, 13https://doi.org/10.3389/fpsyg.2022.1066282

Perez-Gramaje, A. F., Garcia, O. F., Reyes, M., Serra, E., & Garcia, F. (2020). Parenting styles and aggressive adolescents: Relationships with self-esteem and personal maladjustment. European Journal of Psychology Applied to Legal Context, 12(1), 1-10. https://doi.org/10.5093/ejpalc2020a1

Queiroz, P., Garcia, O. F., Garcia, F., Zacares, J. J., & Camino, C. (2020). Self and nature: Parental socialization, self-esteem, and environmental values in Spanish adolescents. International Journal of Environmental Research and Public Health, 17(10), 3732. https://doi.org/10.3390/ijerph17103732

Steinberg, L., Lamborn, S. D., Darling, N., Mounts, N. S., & Dornbusch, S. M. (1994). Over-Time changes in adjustment and competence among adolescents from authoritative, authoritarian, indulgent, and neglectful families. Child Development, 65(3), 754-770. https://doi.org/10.1111/j.1467-8624.1994.tb00781.x

Steinberg, L., Mounts, N. S., Lamborn, S. D., & Dornbusch, S. M. (1991). Authoritative parenting and adolescent adjustment across varied ecological niches. Journal of Research on Adolescence, 1, 19-36.

Villarejo, S., Martinez-Escudero, J. A., & Garcia, O. F. (2020). Parenting styles and their contribution to children personal and social adjustment. Ansiedad y Estrés, 26(1), 1-8. https://doi.org/10.1016/j.anyes.2019.12.001

There are no grerater problems with the quality of English language.

Author Response

Reviewer 3:

  1. We thank the reviewer for their insightful review of the manuscript and have included the following text to the manuscript:

Early parenting research focused on direct means of socialization through parenting styles and practices that describe characteristics of the quality of interactions between parent and child. Initially described by Baumrind (1966) and expanded upon by Maccoby and Martin (1983), this foundational work provides a framework for organizing parenting styles along two orthogonal dimensions of demandingness and responsiveness, often referred to as sensitivity and intrusiveness.

We include the following citations:

Baumrind, D. Effects of authoritative parental control on child behavior Child Dev., 37 (1966), pp. 887-907

Maccoby, E. E., & Martin, J. A. Socialization in the context of the family: parent–child interaction M. Hetherington (Ed.), Handbook of child psychology, Willey (1983)

Garcia, O. F., Fuentes, M. C., Gracia, E., Serra, E., & Garcia, F. (2020). Parenting warmth and strictness across three generations: Parenting styles and psychosocial adjustment. International Journal of Environmental Research and Public Health, 17(20), 7487. https://doi.org/10.3390/ijerph17207487

  1. We have added text to the methods section that higher scores on the self-compassion measure reflect greater self-compassion.

Round 2

Reviewer 2 Report

Not all the issues I raised have been answered. Please read carefully and make the necessary corrections. Corrections must appear in the text

Minor editing of English language required

Author Response

  1. The title has no clear exposition and outcome.

This is not an experimental study but exploratory in nature. We believe the title makes clear that this study is examining factors related to maternal sensitive parenting. As we make clear in the abstract, despite growing research interest linking self-compassion to parenting, there are considerable gaps in the literature. The current study examined the associations between self-compassion, depressive symptoms, socioeconomic status, and sensitive parenting.

  1. Why did you put the title again before the introduction?

We listed title again per APA standards. As we mentioned in the previous response, we will seek guidance from the editor regarding this issue.

  1. References are misplaced in the text. They are not placed after the fullstop. See the author guidelines.

As previously mentioned, this has been corrected.

  1. the «current study» section is not needed.

This has been corrected.

  1. The purpose of your report is not clear.

The purpose of this report is to address an issue in the growing literature linking maternal self-compassion and her parenting. We studied factors that may underpin the association between self-compassion and sensitive parenting. In this particular study, we focused sociodemographic factors and maternal depression, both of which are long proven factors related to mothers’ parenting. 

  1. I also didn’t understand the exclusion criteria. Can you explain the reasons you excluded allergy problems and pregnancies >42 weeks?

Data for this study are drawn from a larger study of gestational age and risk for obesity among young children. Food allergies are natural exclusionary factors when studying metabolic issues. To be clear however, the current study being reviewed is not related to metabolic issues but rather an analysis of self-compassion and maternal caregiving.

  1. Why are measures being developed so much?

To show rigor and transparency of our work.

  1. Table 1 could be omitted as long as the data are reported in the text.

We kept the table in place to allow readers the more clear ‘picture’ of our findings.

  1. The tables are also confusing as they are distributed. It is good that each table with its commentary goes together.

We followed journal guidelines.

  1. The discussion and conclusions are little commented. I think the article needs a complete revision

We have attended to each issue brought forth.

Reviewer 3 Report

The present manuscript requires a thorough theoretical conceptualization. The most important theoretical model within the parental socialization literature (Maccoby and Martin's two-dimensional model) is only named but many of the relevant aspects of this model (dimensions, parental styles and theoretical debate about which parental style is the best) are not explained.

The present manuscript discusses parenting and mentions some of the parenting styles in the introduction without referring to the model that has had the greatest impact on the literature on parenting styles, the two-dimensional theoretical model of Maccoby and Martin. Therefore, the manuscript requires further theoretical conceptualization about the two main parental dimensions, the four parental styles and the theoretical debate about which parental style is the best in different cultural contexts.

Maccoby and Martin's two-dimensional was the model that has had the greatest impact on the parental socialization literature and became the reference model in the parenting literature. This model state that there two main independent parenting dimensions that are used by parents to socialize their children: (i.e., warmth and strictness) (Alcaide et al., 2023; Climent-Galarza et al., 2022; Darling & Steinberg, 1993; Fuentes et al., 2022; F. Garcia & Gracia, 2009; Gimenez-Serrano et al., 2022; Lamborn et al., 1991; Martinez et al., 2020; Martinez-Escudero et al., 2023; Martínez et al., 2021; Palacios et al., 2022; Queiroz et al., 2020).

On the one hand, parental warmth refers to that parents show acceptance,love, affection, emotional closeness, and communication with their children (Alcaide et al., 2023; Climent-Galarza et al., 2022; Lamborn et al., 1991; Martinez et al., 2020; Martinez-Escudero et al., 2020; Martínez et al., 2021). Other labels used to warmth are responsiveness, involvement, implication, security, care or love (Darling & Steinberg, 1993; F. Garcia & Gracia, 2014; Martinez et al., 2020). On the other hand, parental strictness refers to the extent of imposition, authority, or rigidity (Climent-Galarza et al., 2022), the use discipline towards their children, controlling and/or supervising their behavior, establishing norms for children’s behavior, and maintaining position of authority (Alcaide et al., 2023; Darling & Steinberg, 1993).

Maccoby and Martin's two-dimensional model also defends the idea that four parenting styles emerges from the combination of the two main parenting dimensions (i.e., warmth and strictness): authoritarian (strictness but not warmth); authoritative (strictness and warmth), indulgent (warmth but not strictness) and neglectful (neither strictness nor warmth) (Alcaide et al., 2023; Climent-Galarza et al., 2022; Fuentes et al., 2022; Gimenez-Serrano et al., 2022; Martinez-Escudero et al., 2023; Palacios et al., 2022; Perez-Gramaje et al., 2020; Queiroz et al., 2020; Villarejo et al., 2020).

The two-dimensional model gave rise to a theoretical debate about which parenting style is best.

Classical studies about parenting conducted in Anglo-Saxon contexts with European-American samples (mostly white middle-class families) stated that the authoritative parenting (i.e., parental warmth and parental strictness together) was related to the best child psychosocial adjustment (Baumrind, 1991; Lamborn et al., 1991; Steinberg et al., 1991; Steinberg et al., 1994). These results were not always the same in all cultural contexts. Other studies conducted in ethnic minority groups in the United States such as Chinese Americans (Chao, 2001) or African American (Deater-Deckard et al., 1996), and Arabs societies (Dwairy & Achoui, 2006) defended that the authoritarian parenting (i.e., parental strictness without parental warmth) is related to the best child adjustment. Contrary to classical studies, the most recent evidence conducted in European and Latin American countries support the idea that indulgent parenting (i.e, parental warmth without parental strictness) is related to the best child psychosocial adjustment (Alcaide et al., 2023; Fuentes et al., 2022; F. Garcia & Gracia, 2009; O. F. Garcia et al., 2018; O. F. Garcia & Serra, 2019; Gimenez-Serrano et al., 2022; Gimenez-Serrano, Garcia et al., 2022; Martinez-Escudero et al., 2020; Martinez-Escudero et al., 2023; Martínez et al., 2019; Palacios et al., 2022; Perez-Gramaje et al., 2020; Villarejo et al., 2020).

References

Alcaide, M., Garcia, O. F., Queiroz, P., & Garcia, F. (2023). Adjustment and maladjustment to later life: Evidence about early experiences in the family. Frontiers in Psychology, 14https://doi.org/10.3389/fpsyg.2023.1059458

Chao, R. K. (2001). Extending research on the consequences of parenting style for Chinese Americans and European Americans. Child Development, 72, 1832-1843. https://doi.org/10.1111/1467-8624.00381

Climent-Galarza, S., Alcaide, M., Garcia, O. F., Chen, F., & Garcia, F. (2022). Parental socialization, delinquency during adolescence and adjustment in adolescents and adult children. Behavioral Sciences, 12(11)https://doi.org/10.3390/bs12110448

Darling, N., & Steinberg, L. (1993). Parenting style as context: An integrative model. Psychological Bulletin, 113(3), 487-496. https://doi.org/10.1037/0033-2909.113.3.487

Deater-Deckard, K., Dodge, K. A., Bates, J. E., & Pettit, G. S. (1996). Physical discipline among African American and European American mothers: Links to children's externalizing behaviors. Developmental Psychology, 32(6), 1065-1072. https://doi.org/10.1037/0012-1649.32.6.1065

Dwairy, M., & Achoui, M. (2006). Introduction to three cross-regional research studies on parenting styles, individuation, and mental health in Arab societies. Journal of Cross-Cultural Psychology, 37, 221-229. https://doi.org/10.1177/0022022106286921

Fuentes, M. C., Garcia, O. F., Alcaide, M., Garcia-Ros, R., & Garcia, F. (2022). Analyzing when parental warmth but without parental strictness leads to more adolescent empathy and self-concept: Evidence from Spanish homes. Frontiers in Psychology, 13https://doi.org/10.3389/fpsyg.2022.1060821

Garcia, F., & Gracia, E. (2014). The indulgent parenting style and developmental outcomes in South European and Latin American countries. In H. Selin (Ed.), Parenting Across Cultures (pp. 419-433). Springer. https://doi.org/10.1007/978-94-007-7503-9_31

Garcia, F., & Gracia, E. (2009). Is always authoritative the optimum parenting style? Evidence from Spanish families. Adolescence, 44(173), 101-131.

Garcia, O. F., & Serra, E. (2019). Raising children with poor school performance: Parenting styles and short- and long-term consequences for adolescent and adult development. International Journal of Environmental Research and Public Health, 16(7), 1089. https://doi.org/10.3390/ijerph16071089

Garcia, O. F., Serra, E., Zacares, J. J., & Garcia, F. (2018). Parenting styles and short- and long-term socialization outcomes: A study among Spanish adolescents and older adults. Psychosocial Intervention, 27(3), 153-161. https://doi.org/10.5093/pi2018a21

Gimenez-Serrano, S., Alcaide, M., Reyes, M., Zacarés, J. J., & Celdrán, M. (2022). Beyond parenting socialization years: The relationship between parenting dimensions and grandparenting functioning. International Journal of Environmental Research and Public Health, 19(8), 4528. https://doi.org/10.3390/ijerph19084528

Gimenez-Serrano, S., Garcia, F., & Garcia, O. F. (2022). Parenting styles and its relations with personal and social adjustment beyond adolescence: Is the current evidence enough? European Journal of Developmental Psychology, 19(5), 749-769. https://doi.org/10.1080/17405629.2021.1952863

Lamborn, S. D., Mounts, N. S., Steinberg, L., & Dornbusch, S. M. (1991). Patterns of competence and adjustment among adolescents from authoritative, authoritarian, indulgent, and neglectful families. Child Development, 62(5), 1049-1065. https://doi.org/10.1111/j.1467-8624.1991.tb01588.x

Martínez, I., Murgui, S., Garcia, O. F., & Garcia, F. (2021). Parenting and adolescent adjustment: The mediational role of family self-esteem. Journal of Child and Family Studies, 30(5), 1184-1197. https://doi.org/10.1007/s10826-021-01937-z

Martínez, I., Murgui, S., Garcia, O. F., & Garcia, F. (2019). Parenting in the digital era: Protective and risk parenting styles for traditional bullying and cyberbullying victimization. Computers in Human Behavior, 90, 84-92. https://doi.org/10.1016/j.chb.2018.08.036

Martinez, I., Garcia, F., Veiga, F., Garcia, O. F., Rodrigues, Y., & Serra, E. (2020). Parenting styles, internalization of values and self-esteem: A cross-cultural study in Spain, Portugal and Brazil. International Journal of Environmental Research and Public Health, 17(7), 2370. https://doi.org/10.3390/ijerph17072370

Martinez-Escudero, J. A., Garcia, O. F., Alcaide, M., Bochons, I., & Garcia, F. (2023). Parental socialization and adjustment components in adolescents and middle-aged adults: How are they related? Psychology Research and Behavior Management, 16, 1127-1139. https://doi.org/10.2147/PRBM.S394557

Martinez-Escudero, J. A., Villarejo, S., Garcia, O. F., & Garcia, F. (2020). Parental socialization and its impact across the lifespan. Behavioral Sciences, 10(6), 101. https://doi.org/10.3390/bs10060101

Palacios, I., Garcia, O. F., Alcaide, M., & Garcia, F. (2022). Positive parenting style and positive health beyond the authoritative: Self, universalism values, and protection against emotional vulnerability from Spanish adolescents and adult children. Frontiers in Psychology, 13https://doi.org/10.3389/fpsyg.2022.1066282

Perez-Gramaje, A. F., Garcia, O. F., Reyes, M., Serra, E., & Garcia, F. (2020). Parenting styles and aggressive adolescents: Relationships with self-esteem and personal maladjustment. European Journal of Psychology Applied to Legal Context, 12(1), 1-10. https://doi.org/10.5093/ejpalc2020a1

Queiroz, P., Garcia, O. F., Garcia, F., Zacares, J. J., & Camino, C. (2020). Self and nature: Parental socialization, self-esteem, and environmental values in Spanish adolescents. International Journal of Environmental Research and Public Health, 17(10), 3732. https://doi.org/10.3390/ijerph17103732

Steinberg, L., Lamborn, S. D., Darling, N., Mounts, N. S., & Dornbusch, S. M. (1994). Over-Time changes in adjustment and competence among adolescents from authoritative, authoritarian, indulgent, and neglectful families. Child Development, 65(3), 754-770. https://doi.org/10.1111/j.1467-8624.1994.tb00781.x

Steinberg, L., Mounts, N. S., Lamborn, S. D., & Dornbusch, S. M. (1991). Authoritative parenting and adolescent adjustment across varied ecological niches. Journal of Research on Adolescence, 1, 19-36.

Villarejo, S., Martinez-Escudero, J. A., & Garcia, O. F. (2020). Parenting styles and their contribution to children personal and social adjustment. Ansiedad y Estrés, 26(1), 1-8. https://doi.org/10.1016/j.anyes.2019.12.001

The quality of English language is correct.

Author Response

Reviewer 3:

We have edited the manuscript to address issues of dimensions of parenting behavior. The current study is not an analysis of dimensions of parenting. Our efforts were to address gaps in the self-compassion literature and sensitive parenting. We make clear in this revision that future studies should examine the role of self-compassion to other dimension of parenting as this would beyond the scope of this study.